# Association Between the Jiangnan Diet and Mild Cognitive Impairment Among the Elderly

**DOI:** 10.3390/nu17203189

**Published:** 2025-10-10

**Authors:** Mengjie He, Yan Zou, Ronghua Zhang, Danting Su, Peiwei Xu

**Affiliations:** 1Department of Nutrition and Food Safety, Zhejiang Provincial Center for Disease Control and Prevention, Hangzhou 310051, China; mjhe@cdc.zj.cn (M.H.); yzou@cdc.zj.cn (Y.Z.); rhzhang@cdc.zj.cn (R.Z.); dtsu@cdc.zj.cn (D.S.); 2NHC Specialty Laboratory of Food Safety Risk Assessment and Standard Development, Hangzhou 310051, China

**Keywords:** Jiangnan diet, mild cognitive impairment, dietary pattern, elderly

## Abstract

**Background/Objectives**: The Jiangnan diet—a traditional dietary pattern prevalent in Eastern China—is a newly proposed dietary pattern. This study provides additional epidemiological evidence for the promotion of the Jiangnan diet through examining the association between the Jiangnan diet and mild cognitive impairment (MCI). **Methods**: A multicenter cross-sectional study was carried out during 2020 among 1084 community-dwelling adults aged 55 years and above across multiple sites in Zhejiang Province, China. Data collection encompassed basic information of the population, cognition (using the Montreal Cognitive Assessment), dietary intake information (using the Food Frequency Questionnaire, FFQ), life pattern, depressive symptoms (using the Mental Health Assessment Scale for the Elderly), and physical examinations (e.g., height, weight). The dietary patterns were assessed using a validated semi-quantitative FFQ. Factor analysis was used to analyze the 16 categories of food intake of the participants, and dietary patterns and the “Jiangnan diet” were extracted. The Jiangnan diet scores were categorized into quartiles: Q1 (lowest) to Q4 (highest). Multivariate logistic regression was employed to examine the association between adherence to the Jiangnan diet and the prevalence of MCI, with results expressed as odds ratios (OR) and 95% confidence intervals (CI). **Results**: The estimated prevalence of MCI in the study population was 24.6%. The dietary pattern characterized by whole grains, low salt, and low oil was identified as the “Jiangnan diet”. Participants with the highest adherence to the “Jiangnan diet” pattern had 79.2% lower odds of MCI than those with the lowest adherence (odds ratio = 0.208, 95% CI = 0.120~0.362, *p* < 0.0001) after adjusting for age, frequency of social activities, depression, hypertension, alcohol consumption, and energy intake. **Conclusions**: High adherence to the Jiangnan diet was associated with lower odds of MCI. To further verify the relationship between the Jiangnan diet and MCI, future studies will focus on longitudinal research exploring different dietary patterns and disease outcomes across various regions.

## 1. Introduction

Global population aging is accelerating unprecedentedly, with China exemplifying this trend [1,2]. Alzheimer’s disease (AD), the primary cause of dementia in aging societies, imposes a severe burden on households, national healthcare infrastructures, and the global community [3]. An estimated 55 million people globally live with AD, around 10 million of whom are in China [4,5]. Despite decades of research, limited AD treatment trials have succeeded to date [6], underscoring the critical importance of primary prevention. Mild cognitive impairment (MCI), a transitional phase between normal cognitive decline and dementia, progresses to dementia at an annual rate of 10–20% [7]. Identifying high-risk populations and modifiable protective factors for MCI is therefore essential to developing effective and targeted strategies for preventing AD.

Lifestyle interventions targeting dietary modifications represent a key strategy for significantly reducing the risk of chronic diseases [8]. The Mediterranean-DASH Intervention for Neurodegenerative Delay (MIND) diet was designed for neuroprotection and is associated with reduced all-cause dementia risk and slower cognitive decline in prospective studies [9,10]. The Jiangnan diet—a historically rooted eating habit among inhabitants of the Yangtze River Delta region for centuries—represents a promising candidate for future national dietary recommendations in China, analogous to the MIND diet in Western countries. The Jiangnan dietary pattern—representative of the Oriental healthy dietary model in the Chinese Dietary Guidelines 2022 [11]—is characterized by light seasoning, nutritional balance, and culinary refinement. Core components are as follows: (1) high intake of vegetables and fruits to achieve a dietary pattern of low sodium and high potassium; (2) predominance of whole grains, which replace refined cereals; (3) rapeseed oil as the main lipid source, given its richness in monounsaturated and polyunsaturated fatty acids; and (4) a diet primarily based on white meat, with substantial intake of aquatic products (see Figure 1). Some studies suggest that the Jiangnan diet may help prevent and control cardiovascular diseases (CVDs) [12,13] There is also research indicating that it might contribute to preventing the occurrence of sarcopenia [14]. However, current evidence on the relationship between the Jiangnan diet and cognitive impairment remains extremely scarce. Evidence suggests that B vitamins rich in whole grains, vitamin C in fruits, and ω-3 fatty acids abundant in seafood contribute significantly to cognitive preservation [15,16,17,18,19]. Therefore, the core characteristics of the Jiangnan diet may serve as protective factors for cognitive function. Notably, epidemiological surveillance data indicate that Zhejiang Province exhibits a standardized AD incidence rate lower than the national average [20,21], which suggests a potential neuroprotective effect of the historically preserved Jiangnan dietary pattern prevalent in this region.

The characteristics of the Jiangnan diet demonstrate concordance with key neuroprotective factors against AD pathogenesis. However, limited research has comprehensively examined the relationship between adherence to the Jiangnan diet and the prevalence of MCI. This study intended to investigate the relationship between the Jiangnan diet and MCI prevalence, with the goal of informing feasible dietary strategies for preventing cognitive decline at the pre-dementia stage of Alzheimer’s disease within the Chinese population.

## 2. Materials and Methods

### 2.1. Study Protocol

Our study utilized the follow-up evaluation data from the community-based cohort study on nervous system diseases—the Alzheimer’s disease cohort [22], a prospective investigation initiated in 2018 to identify modifiable risk factors for AD in Chinese adults aged 55 and above, with Zhejiang Province serving as one of its four national survey sites. Detailed cohort methodology has been published previously. In September 2020, a follow-up field survey of community-dwelling participants was conducted across multiple sites in Zhejiang Province, China.

The study protocol received approval from the Research Ethics Committee of the National Institute for Nutrition and Health, Chinese Center for Disease Control and Prevention (Approval No. 2017020; 6 November 2017). This study was conducted following the ethical principles of the Declaration of Helsinki. All participants provided written informed consent before enrollment.

### 2.2. Target Population and Sampling Strategy

The baseline cohort was constructed by employing a multistage, stratified probability sampling method, encompassing participants from multiple stages throughout Zhejiang Province, China [15].

Enrollment criteria encompassed the following:Age ≥ 55 years;Permanent residency (defined as ≥5 years’ residency) amidst the target community;Nonexistence of comorbidities potentially confounding cognitive assessments, including congenital/acquired intellectual disability, severe psychiatric disorders (e.g., schizophrenia or bipolar disorder), or uncorrectable visual/auditory impairments.

Following rigorous quality control procedures—standardized protocols for data verification and electronic consistency checks—a final analytical sample of 1084 participants was enrolled, with detailed participant flow summarized in Figure 2.

### 2.3. Data Gathering

Data gathering comprised the following: (1) Structured questionnaires assessed cognitive function (using the Mental Health Assessment Scale for the Elderly), dietary intake via Food Frequency Questionnaire (FFQ), sociodemographic factors (region, gender, age, education level, occupation, marital status), and health-influencing factors (chronic disease prevalence, smoking, alcohol consumption, sleep, depressive symptoms, social activities, etc.); (2) Standardized physical examinations by certified health professionals comprising morphological measurements (height, weight; body mass index (BMI) derived as kg/m^2^). Quality control involved pre-survey instrument calibration and in-survey electronic data validation [15].

### 2.4. Cognitive Function Evaluation

The Montreal Cognitive Assessment (MoCA) was utilized to evaluate the cognitive function, which was selected for its superior sensitivity to MCI compared to the Mini-Mental State Examination (MMSE) [23]. The Beijing MoCA version [24,25,26]—validated in Chinese populations—was administered via standardized face-to-face interviews by trained investigators. MCI was diagnosed per established Chinese normative criteria [24]: ≤13 for illiterate people, ≤19 for those with 1–6 years of education, and ≤24 for those with 7 or more years of education.

### 2.5. Dietary Pattern and Energy Evaluation

Dietary patterns were assessed using FFQ (containing 64 categories of foods) and converted FFQ Food Composition Tables (FCTs) [27]. A total of 64 categories of foods were classified into 10 food groups according to the Dietary Pagoda, among which tofu, dried tofu skin, and soybean milk were converted into dried soybeans based on their protein proportion, and grouping criteria are detailed in Table 1. Factor analysis on participants’ food intake across 10 categories was performed to derive dietary patterns. Sampling adequacy was confirmed by a Kaiser–Meyer–Olkin (KMO) statistic > 0.6 and Bartlett’s test of sphericity (*p* < 0.05). Common factors were retained based on eigenvalues > 1 and scree plot inflection points. The orthogonal varimax rotation method assigned factor loadings to each food group, with |loading| ≥ 0.3 defining dominant variables per dietary pattern. Regression-derived factor scores quantified individual adherence: positive scores indicated higher consumption of pattern-specific foods, negative scores indicated lower consumption, and higher absolute scores reflected stronger pattern adherence. Dietary pattern scores were categorized into quartiles (Q1–Q4). Building upon the China Health and Nutrition Survey (CHNS) data [28] and the Weighted Food Composition Table (WFCT) methodology [29], our research team developed specialized FFQ FCTs [27]. These FFQ FCTs were subsequently utilized to calculate daily per capita energy intake based on dietary data collected via the study-specific FFQ.

### 2.6. Assessment of Covariates

Previous studies indicated that overweight and obesity may be associated with cognitive impairment [30]. According to the BMI criteria, a value of ≥24.0 kg/m^2^ is defined as overweight or obesity. Therefore, participants were categorized by BMI into two groups: <24.0 kg/m^2^ and ≥24.0 kg/m^2^ [31]. Depressive symptoms were evaluated with the Geriatric Depression Scale (GDS) [32]. Sleep disturbance was evaluated using the Chinese-validated Pittsburgh Sleep Quality Index (PSQI), with a score > 5 indicating poor sleep quality [33,34]. Smoking and alcohol consumption were recorded dichotomously (yes/no) based on self-reported history of use. Hypertension was defined as a prior diagnosis or an average of three physical exam measurements exceeding 140 mmHg systolic or 90 mmHg diastolic pressure [35].

### 2.7. Statistical Analyses

The Kolmogorov–Smirnov test was performed to test the normality of the intake of various food categories. Continuous variables with a normal distribution were described using mean and standard deviation (mean ± SD), while those with a skewed distribution were summarized with median and quartiles [median (Q1, Q3)]. Categorical variables were expressed as frequency and percentage (%) and compared by the chi-square test. Dietary patterns were derived using principal component analysis (PCA). The association between the Jiangnan diet and MCI prevalence was examined through multiple logistic regression, with results presented as adjusted odds ratios (ORs) and 95% confidence intervals (CIs). Potential confounders, including gender, age, education level, occupation, marital status, BMI, social activity frequency, depression, sleep disturbances, alcohol consumption, smoking, energy intake, diabetes, and hypertension, were included in the multivariate models. All analyses were conducted with SAS 9.1 and SPSS 21.0. A two-sided *p*-value < 0.05 was considered statistically significant.

## 3. Results

### 3.1. Establishment of Dietary Pattern

The KMO value was 0.842, and Bartlett’s test of sphericity reached statistical significance (*p* < 0.0001), confirming the suitability of the data for dietary pattern analysis. Four distinct dietary patterns were identified, each demonstrating specific factor loadings and proportions of explained variance (Table 2). Four major dietary patterns were identified, derived from the ten food groups that significantly contributed to each pattern in the KMO test. The first was named the “animal products” pattern, which mainly contained freshwater aquatic products, poultry and livestock meat, nuts, and fruits. The second one, the “whole Grains-tubers” pattern, had a unique character with a higher consumption of whole grains and miscellaneous tubers, vegetables, soybeans, and soybean products. Within the third pattern, the “High in salt and oil” pattern, participants within this pattern preferred a diet high in oil and salt. The last one was overrepresented by wheat and wheat products; therefore, it was labeled as the “Wheat and wheat products” pattern. The variance for these four patterns was 33.27%, 15.05%, 7.68%, and 6.07%, respectively. Collectively, they explained 62.07% of the variance in the primary variable. In the “whole Grains-tubers” dietary pattern, whole grains and miscellaneous tubers, vegetables, soybeans, and soybean products ranked among the top positive contributors; meanwhile, oil and salt showed negative loadings, indicating low consumption of oil and salt within this pattern. This aligns well with the characteristics of the Jiangnan diet, so the “whole Grains-tubers” dietary pattern was defined as the “Jiangnan diet”.

### 3.2. Basic Information

This study included 1084 participants, with key characteristics summarized in Table 3. The overall prevalence of MCI was 24.6%. The sample was well distributed across regions, genders, ages. Normality tests revealed that neither the intake of individual food categories nor the dietary pattern scores conformed to a normal distribution (*p* < 0.01); thus, the quartile method was employed for subsequent analyses. A significant inverse trend between higher Jiangnan diet scores and lower MCI prevalence was observed (Cochran–Mantel–Haenszel chi-square test, *p* < 0.05).

### 3.3. Dietary Pattern and Its Association with Mild Cognitive Impairment

Table 1 presents the variables included in the univariate chi-square analysis, which covered dietary pattern scores, sociodemographic factors (gender, age, education, occupation, marital status), health-related indicators (BMI, social activity frequency, depression, sleep disturbances, smoking, alcohol consumption, diabetes, energy intake, hypertension), and others. A significant upward trend in MCI prevalence was observed with increasing Jiangnan diet adherence (*p* < 0.05). Multivariate analysis revealed that, compared to the lowest adherence quartile (Q1), higher Jiangnan diet scores were associated with significantly reduced prevalence of MCI: Q2 (OR = 0.53, 95% CI: 0.36–0.80), Q3 (OR = 0.32, 95% CI: 0.21–0.50), and Q4 (OR = 0.21, 95% CI: 0.12–0.36), after adjusting for age, social activity frequency, depression, hypertension, alcohol consumption, and energy intake (Figure 3). These results remained consistent after additional threshold adjustments (Appendix A).

## 4. Discussion

Given the limited effective therapies for AD, implementing multifaceted primary prevention approaches to detect and manage MCI before its progression to dementia remains a critical public health imperative. This study examined the association between adherence to the Jiangnan diet and MCI prevalence to explore a suitable MIND dietary pattern for the Chinese population. In this study, the analysis revealed four major dietary patterns, in which the “Whole Grains-tubers” dietary pattern was defined as the “Jiangnan diet”. Multivariate analysis revealed that those who adhered most strongly to the “Jiangnan diet” pattern had 79.2% lower odds of MCI than those with the lowest adherence (OR = 0.208, 95% CI = 0.120~0.362, *p* < 0.0001) after adjusting for age, frequency of social activities, depression, hypertension, alcohol consumption, and energy intake. Additionally, a significant inverse association was found between Jiangnan diet scores and the prevalence of MCI.

The China Health and Retirement Longitudinal Study, which assessed 3622 older adults using the MMSE scale, reported a national cognitive impairment prevalence of 41.6% [36]. However, the prevalence of MCI in Zhejiang, a region where the Jiangnan diet is commonly followed, was only 24.6%. The observed inverse association between regional MCI prevalence and adherence to the Jiangnan diet suggests its potential protective effect against cognitive decline, which may partly explain the geographic variation in MCI rates across China.

In this study, we derived four dietary patterns through factor analysis. By observing the food groups with high factor loadings across these patterns, it was found that the “Animal and aquatic products” pattern and the “Whole grains” pattern most closely align with the key features of the Jiangnan dietary pattern. Within the “Whole grains” pattern, oil and salt exhibited negative loadings, indicating lower consumption of these components, whereas the “Animal and aquatic products” pattern showed a positive loading for salt. Based on these characteristics, the “Whole grains” pattern was identified as representative of the Jiangnan diet [37]. Subsequent correlation analysis will directly explore the association between this dietary pattern and MCI.

The Yangtze River Delta region, which includes Zhejiang Province, is rich in rapeseed oil, and residents in this region prefer to use rapeseed oil for cooking.

Several studies have indicated that rapeseed oil, which contains a balanced profile of ω-6 and ω-3 fatty acids along with an optimal ω-6/ω-3 ratio, may be associated with a reduced risk of cognitive decline and dementia [38,39]. The Yangtze River Delta Region obtains its name from the delta formed at the estuary of the Yangtze River [40]. Coupled with freshwater lakes such as Taihu Lake, this geographical feature ensures high accessibility to both seafood and freshwater products for the region’s residents. The results indicate that aquatic products rank first as the principal component in the animal and aquatic products” pattern, which proves that residents in this region consume a relatively high amount of aquatic products. Based on existing research, aquatic products, especially deep-sea fish, are rich in omega-3 fatty acids, which may act as a protective factor against cognitive impairment [18,19,41]. The PCA results also indicated that whole grains rank first. A number of studies have established that B vitamins, abundantly present in whole grains, play a protective role in cognitive function [42,43]. Furthermore, fruits and vegetables consistently emerge as significant components in principal component analysis, with their richness in dietary fiber, vitamin C, and other essential nutrients being important protective factors for cognition [44,45]. The characteristics of the Jiangnan diet practiced in Zhejiang Province include a stronger preference for cooking methods such as blanching and steaming. A low-salt diet can not only reduce the risk of chronic diseases such as CVD but also potentially improve cognitive function [46,47]. Although there are currently few research reports on the relationship between the Jiangnan diet and cognitive function, the aforementioned studies on the association between the components of the Jiangnan diet and cognition suggest that the Jiangnan diet may be a version of the MIND diet that is suitable for Chinese residents.

Covariate analysis revealed that advanced age, depression, hypertension diagnosis, and limited social engagement showed a significant correlation with an increased prevalence of MCI. These findings align with established risk factors documented in prior studies, underscoring the robustness and validity of the current dataset [48,49,50,51,52]. In univariate analyses, both educational level and sleep disturbances exhibited significant correlations with MCI; however, these associations were not retained in the multivariate model. The observed attenuation may be attributed to multicollinearity, particularly considering the strong correlation between depression and sleep disturbances [53]. Although variables such as educational attainment, BMI, marital status, and smoking history have been previously implicated in MCI risk [54,55,56], they did not emerge as significant predictors in our study.

This research is characterized by several key advantages. The first one, Zhejiang Province, due to its unique geographical location, is highly suitable for conducting research on the Jiangnan diet. The other advantage of this study is that, as far as we are aware, it is the first to explore the link between the Jiangnan diet and the prevalence of MCI in a Chinese population.

This research has several limitations that should be considered when interpreting the results. First, the cross-sectional design precludes the establishment of causal relationships between dietary pattern and MCI, limiting insight into temporal dynamics. Second, the use of self-reported dietary data may introduce recall bias and measurement inaccuracy. Third, the geographic scope was confined to Zhejiang Province rather than encompassing the entire Yangtze River Delta region, which somewhat limits the representativeness of the Jiangnan diet examined. Finally, owing to the absence of supporting imaging diagnostic data in this study, we were unable to determine whether the MCI was attributed to AD or vascular etiologies, nor could we further subclassify MCI.

Future investigations would benefit from employing longitudinal designs with larger sample sizes and survey areas, which are necessary to corroborate these observations and explore their long-term implications. Such studies would help delineate causal pathways and reduce measurement bias through more objective dietary assessment methods. Additionally, expanding the sampling to include diverse populations across the Yangtze River Delta would enhance the generalizability and ecological validity of the results.

## 5. Conclusions

In summary, participants with the highest adherence to the “Jiangnan diet” pattern had 79.2% lower odds of MCI than those with the lowest adherence, suggesting that Jiangnan diet adherence is associated with reduced MCI prevalence, thus providing scientific evidence for the further promotion of the Jiangnan diet. Future large-scale longitudinal studies are needed to establish causality and develop targeted preventive strategies.

## Figures and Tables

**Figure 1 nutrients-17-03189-f001:**
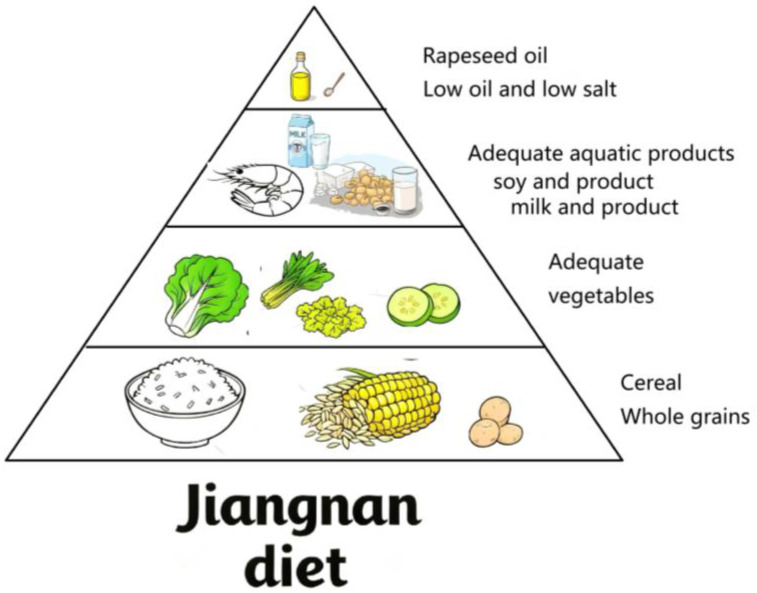
The characteristics of the Jiangnan diet.

**Figure 2 nutrients-17-03189-f002:**
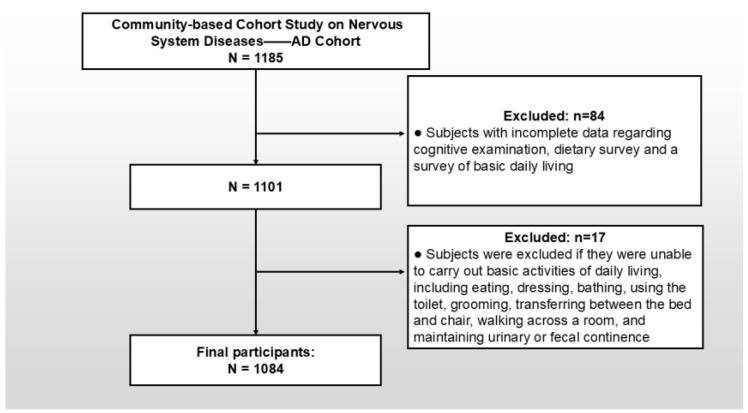
Illustrates the derivation of the analytical sample. The direction of the arrow indicates the order of inclusion and exclusion. AD: Alzheimer’s disease.

**Figure 3 nutrients-17-03189-f003:**
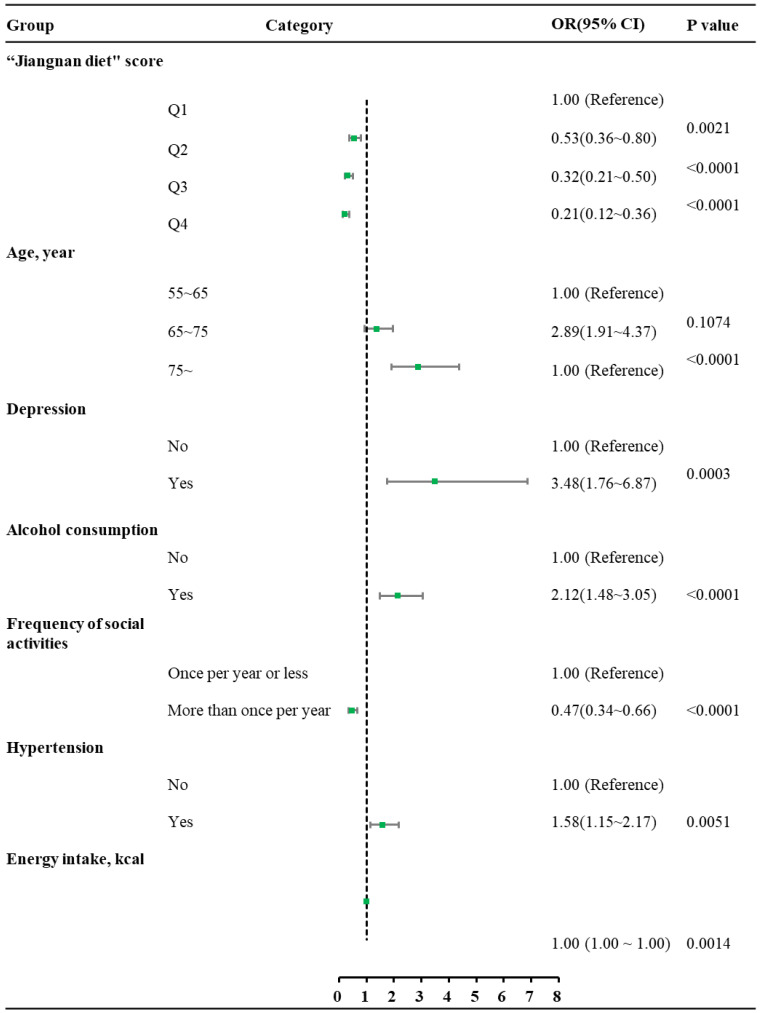
Forest plot of MCI prevalence from multivariate logistic regression. Green squares represent OR, and horizontal lines depict 95% CI. OR: Odds ratios; CI: Confidence intervals.

**Table 1 nutrients-17-03189-t001:** Classification of foods for dietary pattern analysis.

Food Groups	Food Items
Rice and rice products	Rice, rice flour, etc.
Wheat and wheat products	Bread, steamed buns, noodles, dumplings, etc.
Whole grains and miscellaneous	Whole-wheat bread, buckwheat, corn, millet, sorghum, coix seed, rye, mung beans, adzuki beans, pinto beans, kidney beans, etc.
Tubers	Sweet potatoes, potatoes, taros, yams, etc.
Soybeans and soybean products	Dried soybeans (soybeans, green soybeans, or black soybeans), soy milk, soybean flour, tofu, tofu skin, etc.
Vegetables	Fresh legumes, tomatoes, peppers, carrots, melons, vegetables, leafy greens, Chinese cabbage and other leafy vegetables, cruciferous vegetables, other fresh or frozen vegetables, green onions and garlic, fungi and algae, dried vegetables
Fruits	Orange fruits, watermelons, melons, other melon fruits, berry fruits, figs, all other fresh/frozen fruits, all other dried fruits except preserved fruits
Milk and milk products	Whole liquid milk, low-fat liquid milk/skim liquid milk, whole milk powder, low-fat milk powder/skim milk powder, yogurt, cheese, ice cream
Meat	Chicken, duck, goose, pigeon, quail, meat, lean pork, fatty pork, beef, lamb, other non-processed meat, processed meat products
Seafood	Marine fish, marine shrimp and crabs, and other seafood
Freshwater aquatic products	Freshwater fish, freshwater shrimps, and crabs
Eggs	Fresh eggs (eggs, duck eggs, goose eggs, quail eggs), salted eggs, salted duck eggs, salted goose eggs, pine eggs
Nuts	Peanuts, melon seeds, pumpkin seeds, seaweeds, melon seeds, other melon seeds
Sault	Sault
Oil	Vegetable oil and animal oil
Vegetable oil	Rapeseed oil, peanut oil, corn oil, rice bran oil

**Table 2 nutrients-17-03189-t002:** Dietary pattern factor loadings.

Dietary Pattern	Influence Coefficient	Root Test Value	% of Variance	Cumulative%
“Animal products” pattern		5.32	33.272	33.272
Freshwater aquatic products	0.712			
Meat	0.688			
Nuts	0.654			
Fruits	0.648			
Oil	−0.074			
Vegetable oil	−0.060			
Sault	0.100			
“Whole Grains-tubers” pattern		2.41	15.052	48.324
Whole grains and miscellaneous	0.808			
Tubers	0.775			
Vegetables	0.685			
Soybeans and soybean products	0.615			
Oil	−0.029			
Vegetable oil	−0.022			
Sault	−0.063			
“High in salt and oil” pattern		1.23	7.677	56.001
Oil	0.956			
Vegetable oil	0.955			
Sault	0.755			
“Wheat-based foods” pattern		0.97	6.070	62.071
Wheat and wheat products	0.616			

**Table 3 nutrients-17-03189-t003:** Demographic characteristics of the study sample.

Variables	Total, *N*	MCI, *N* (%)	Normal, *N* (%)	χ^2^	*p*
Total	1084	267 (24.6)	817 (75.4)		
“Animal products” pattern				**27.09**	**<0.0001**
Q1	276 (25.5)	93 (34.8)	183 (22.4)		
Q2	269 (24.8)	76 (28.5)	193 (23.6)		
Q3	268 (24.7)	44 (16.5)	224 (27.4)		
Q4	271 (25)	54 (20.2)	217 (26.6)		
“Whole Grains-tubers” pattern				**24.8** **6**	**<0.0001**
Q1	276 (25.5)	95 (35.6)	181 (22.2)		
Q2	269 (24.8)	71 (26.6)	198 (24.2)		
Q3	269 (24.8)	49 (18.4)	220 (26.9)		
Q4	270 (24.9)	52 (19.5)	218 (26.7)		
“High in salt and oil” pattern				**33.5** **3**	**<0.0001**
Q1	277 (25.6)	87 (32.6)	190 (23.3)		
Q2	268 (24.7)	84 (31.5)	184 (22.5)		
Q3	270 (24.9)	61 (22.9)	209 (25.6)		
Q4	269 (24.8)	35 (13.1)	234 (28.6)		
“Wheat-based foods” pattern				**27.18**	**<0.0001**
Q1	276 (25.5)	58 (21.7)	218 (26.7)		
Q2	270 (24.9)	42 (15.7)	228 (27.9)		
Q3	268 (24.7)	89 (33.3)	179 (21.9)		
Q4	270 (24.9)	78 (29.2)	192 (23.5)		
Region				3.13	0.077
Urban	554 (51.1)	149 (55.8)	405 (49.6)		
Rural	530 (48.9)	118 (44.2)	412 (50.4)		
Gender				0.01	0.932
Male	514 (47.4)	126 (47.2)	388 (47.5)		
Female	570 (52.6)	141 (52.8)	429 (52.5)		
Age, year				**43.57**	**<0.001**
55~65	372 (34.3)	63 (23.6)	309 (37.8)		
65~75	479 (44.2)	110 (41.2)	369 (45.2)		
75~	233 (21.5)	94 (35.2)	139 (17.0)		
Educational level				**4.6**	**0.032**
Junior school or below	927 (86.5)	236 (90.4)	691 (85.2)		
Senior high school or above	145 (13.5)	25 (9.6)	120 (14.8)		
Occupation				0.14	0.710
Employed or re-employed after retirement or seeking employment	166 (15.3)	39 (14.6)	127 (15.5)		
Retired or unemployed	918 (84.7)	228 (85.4)	690 (84.5)		
Marriage				1.11	0.293
Married	919 (84.8)	221 (82.8)	698 (85.4)		
Unmarried/divorced/widowed	165 (15.2)	46 (17.2)	119 (14.6)		
BMI (kg/m^2^)				0.46	0.498
<24.0	614 (56.6)	156 (58.4)	458 (56.1)		
≥24	470 (43.4)	111 (41.6)	359 (43.9)		
Frequency of social activities				**24.04**	**<0.001**
Once per year or less	275 (25.4)	98 (36.7)	177 (21.7)		
More than once per year	809 (74.6)	169 (63.3)	640 (78.3)		
Depression				**20.09**	**<0.001**
No	1041 (96.0)	244 (91.4)	797 (97.6)		
Yes	43 (4.0)	23 (8.6)	20 (2.5)		
Sleep disturbances				**6.10**	**0.014**
No	585 (54.1)	126 (47.6)	459 (56.3)		
Yes	496 (45.9)	139 (52.5)	357 (43.8)		
Smoking				2.58	0.108
No	882 (81.5)	208 (78.2)	674 (82.6)		
Yes	200 (18.5)	58 (21.8)	142 (17.4)		
Alcohol consumption				**10.89**	**0.012**
No	874 (80.6)	197 (73.8)	677 (82.9)		
Yes	210(19.4)	70(26.2)	140(17.1)		
Hypertension				**15.58**	**<0.001**
No	499 (46.0)	95 (35.6)	404 (49.5)		
Yes	585 (54.0)	172 (64.4)	413 (50.6)		
Diabetes				2.71	0.100
No	988 (91.6)	238 (89.1)	750 (92.4)		
Yes	91 (8.4)	29 (10.9)	62 (7.6)		
Energy intake, kcal *	1517.5 ± 1282.4	1528.8 ± 1495	1513.8 ± 1205.8	−0.15	0.882

* Data are presented as mean ± SD and analyzed by *t*-test. Missing values were as follows: education level (*n* = 12), occupation (*n* = 12), BMI (*n* = 32), sleep disturbances (*n* = 3), smoking (*n* = 2), and diabetes (*n* = 5). Statistically significant results are highlighted in bold. MCI: Mild Cognitive Impairment.

## Data Availability

The datasets and analytic code generated during this study are not publicly available due to privacy and ethical restrictions. However, detailed methodologies regarding dietary and MCI assessment are available from the corresponding author upon reasonable request: pwxu@cdc.zj.cn.

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
