# Peer review of "Association Between the Jiangnan Diet and Mild Cognitive Impairment Among the Elderly"

_nutrients, 2025, doi:10.3390/nu17203189_

Round 1

Reviewer 1 Report

Comments and Suggestions for Authors

You have a very high similarity index, which supposes a lot of coincidences with other published works. For one paper only, you have 27% of similarities, which is unacceptable.

This is the list of the papers with more coincidences with your manuscript:

https://www.mdpi.com/2072-6643/17/13/2111

https://www.frontiersin.org/journals/nutrition/articles/10.3389/fnut.2025.1535226/full

https://www.mdpi.com/2072-6643/13/4/1341/xml

https://www.mdpi.com/2072-6643/17/2/248

In the abstract, you should indicate some directions for further investigations, future perspectives, and practical implications.

More data on the Jiangnan diet patterns should be provided in the introductory section, as well as some illustrations. Further justifications for conducting this work and its relevance for the international community (scientific and population) should be given.

Include more details regarding the assessment of cognitive function in section 2.4.

I suggest the merging of Results with Discussion. Further analysis and discussion on your study’s limitations should be addressed, as well as the study’s strengths.

The Conclusions are poor; please work on this section and make it clear how your results are relevant. Practical implications and future perspectives need to be proposed.

Author Response

Reviewer 1:

Comment 1: You have a very high similarity index, which supposes a lot of coincidences with other published works. For one paper only, you have 27% of similarities, which is unacceptable.

This is the list of the papers with more coincidences with your manuscript:

https://www.mdpi.com/2072-6643/17/13/2111

https://www.frontiersin.org/journals/nutrition/articles/10.3389/fnut.2025.1535226/full

https://www.mdpi.com/2072-6643/13/4/1341/xml

https://www.mdpi.com/2072-6643/17/2/248

Response: Thank you for your helpful comments. The revisions have been made in the manuscript to reduce the similarity rate.

Comment 2: In the abstract, you should indicate some directions for further investigations, future perspectives, and practical implications.

Response: Further investigations, future perspectives, and practical implications have added in the conclusion. The background section in the abstract has been revised to include the significance of studying the association between the Jiangnan diet and mild cognitive impairment (MCI).

Comment 3: More data on the Jiangnan diet patterns should be provided in the introductory section, as well as some illustrations. Further justifications for conducting this work and its relevance for the international community (scientific and population) should be given.

Response: The illustration of the characteristics of Jiangnan diet has been added to the introduction. Since the Jiangnan dietary pattern was first proposed in 2022, relevant epidemiological evidence remains insufficient. The existing studies on its association with cardiovascular diseases have been included in this article.

Comment 4: Include more details regarding the assessment of cognitive function in section 2.4.

Response: More details regarding the assessment of cognitive function such as diagnosed criteria was added in section 2.4.

Comment 5: I suggest the merging of Results with Discussion. Further analysis and discussion on your study’s limitations should be addressed, as well as the study’s strengths.

Response: Revisions have been made in the manuscript: the results have been fully integrated into the discussion section, with comparative analyses conducted. Strengths along its limitations of this study, have been addressed.

Comment 6:

The Conclusions are poor; please work on this section and make it clear how your results are relevant. Practical implications and future perspectives need to be proposed.

Response: Conclusions was revised in the manuscript.

Reviewer 2 Report

Comments and Suggestions for Authors

This is an interesting study, confirming the protective role of a diet based on whole-grain carbohydrates, a low consumption of animal protein, and a higher intake of plant-based protein, as well as a reduced intake of fat, on the incidence of MCI.
However, the authors could make some small changes to make the manuscript more interesting:
1. The authors should specify the test used to determine whether and which variables follow a normal distribution (Kolmogorov-Smirnov test or other).
2. The authors could develop a food pyramid based on the Jiangnan dietary model, modeled after the Mediterranean diet.
3. The authors could summarize the percentage of food intake (intake of carbohydrates from wheat, protein from animal meat, protein from soy, etc.) in a table or graph.

Author Response

Reviewer 2:

Comment 1: This is an interesting study, confirming the protective role of a diet based on whole-grain carbohydrates, a low consumption of animal protein, and a higher intake of plant-based protein, as well as a reduced intake of fat, on the incidence of MCI.

Response: Thank you very much for your positive feedback on our study. Your recognition of our research is deeply encouraging to our research team.
Comment 2: However, the authors could make some small changes to make the manuscript more interesting:
1. The authors should specify the test used to determine whether and which variables follow a normal distribution (Kolmogorov-Smirnov test or other).
2. The authors could develop a food pyramid based on the Jiangnan dietary model, modeled after the Mediterranean diet.
3. The authors could summarize the percentage of food intake (intake of carbohydrates from wheat, protein from animal meat, protein from soy, etc.) in a table or graph

Response: Thank you very much for your suggestions. Normal distribution test has been added. A food pyramid based on the Jiangnan dietary model has been depicted. Since this study adopted the Food Frequency Questionnaire (FFQ) method for dietary survey, which collected the intake data of 64 food categories, the carbohydrate content varies across different categories. Calculating the macronutrient content of each food category would easily lead to significant errors; therefore, no corresponding table has been provided.

Reviewer 3 Report

Comments and Suggestions for Authors

Ref.: 3883907

This is an interesting cross-sectional study, examining the possible association between the Jiangnan diet (a traditional Chinese diet) and the prevalence of mild cognitive impairment (MCI). This diet encompasses low sodium and high potassium intake, whole grains, rapeseed oil, white meat and abundant aquatic product intake. It may be considered as a Chinese analogue of the Mediterranean diet.  The authors observed that high adherence to the Jiangnan diet was associated with lower odds of MCI. There are many risk factors for cognitive impairment and among them dietary factors may be important since they are modifiable. Thus, such studies, investigating the possible preventing role of some types of diet are very welcomed.

The study was well designed and executed. The number of participants is adequate. Statistics are appropriate and well presented. References are up to date. The authors describe the strengths and weaknesses of the study and provide future directions.

Here are some points:

  • In Figure 1, one could get the impression the subjects excluded due to inability to curry out basic activities of daily living (n=17) had moderate/severe dementia. Then, subjects with mild dementia were included or excluded? Please clarify.
  • 1st line of Discussion: “In the absence of disease-modifying therapies for Alzheimer's disease…”. Please avoid such a statement. The anti-amyloid antibodies, such as lecanemab or donanemab may be considered as disease modifying treatments.
  • Could the Jiangnan diet be effective in the prevention of Alzheimer’s disease only, or of other causes of cognitive impairment as well (such as vascular cognitive impairment)? Please comment on that (1-2 sentences in the discussion would be adequate).
  • Are there any clinical/imaging/biomarker data allowing to subdivide MCI? Amnestic, frontal, multidomain, due to Alzheimer’s disease or to non-Alzheimer’s disorder? If such data are not available, please add this in the limitations of the study.

Author Response

Reviewer 3:

Here are some points:

Comment 1: This is an interesting cross-sectional study, examining the possible association between the Jiangnan diet (a traditional Chinese diet) and the prevalence of mild cognitive impairment (MCI). This diet encompasses low sodium and high potassium intake, whole grains, rapeseed oil, white meat and abundant aquatic product intake. It may be considered as a Chinese analogue of the Mediterranean diet.  The authors observed that high adherence to the Jiangnan diet was associated with lower odds of MCI. There are many risk factors for cognitive impairment and among them dietary factors may be important since they are modifiable. Thus, such studies, investigating the possible preventing role of some types of diet are very welcomed.

The study was well designed and executed. The number of participants is adequate. Statistics are appropriate and well presented. References are up to date. The authors describe the strengths and weaknesses of the study and provide future directions.

Response: Thank you very much for your positive feedback on our study. Your recognition of our research is deeply encouraging to our research team.
Comment 2: •     In Figure 1, one could get the impression the subjects excluded due to inability to curry out basic activities of daily living (n=17) had moderate/severe dementia. Then, subjects with mild dementia were included or excluded? Please clarify.
Response: Thank you very much for your suggestions. Based on the definition of MCI, subjects were excluded if they were unable to carry out basic activities of daily living, including eating, dressing, bathing, using the toilet, grooming, transferring between the bed and chair, walking across a room, and main-taining urinary or fecal continence (n = 17). Subjects with mild dementia were not excluded.

Comment 3: •1st line of Discussion: “In the absence of disease-modifying therapies for Alzheimer's disease…”. Please avoid such a statement. The anti-amyloid antibodies, such as lecanemab or donanemab may be considered as disease modifying treatments.

Response: Thank you very much for your suggestions. The statement in the manuscript has been revised from "the absence of " to "limited effective therapies for AD ".

Comment 4: Could the Jiangnan diet be effective in the prevention of Alzheimer’s disease only, or of other causes of cognitive impairment as well (such as vascular cognitive impairment)? Please comment on that (1-2 sentences in the discussion would be adequate).

Response: Thank you very much for your suggestions. The revisions have been supplemented in the "Limitations" section of the discussion.

Comment 5: Are there any clinical/imaging/biomarker data allowing to subdivide MCI? Amnestic, frontal, multidomain, due to Alzheimer’s disease or to non-Alzheimer’s disorder? If such data are not available, please add this in the limitations of the study.

Response: Thank you very much for your suggestions. The revisions have been supplemented in the "Limitations" section of the discussion.

Reviewer 4 Report

Comments and Suggestions for Authors

In the work ´Association between the Jiangnan diet and mild cognitive impairment among the elderly´, the association between the adherence to the Jiangnan diet (a traditional dietary pattern in Eastern China) and the prevalence of MCI was examined via a cross-sectional study enrolling 1084 adults aged 55 years and above from multiple sites in Zhejiang Province (China). The manuscript can be recommended for publication after addressing some minor remarks.

As the authors stated in the limitations of the work, spite the population is not fully representative of the overall region with this diet pattern, the amount of individuals evaluated allow to provide solid conclusions. In this context, the conclusions of the work must be further elaborated.

The section Study protocol must be further detailed/clarified. ´Our study utilized the follow-up evaluation data from the Community-based Cohort Study on Nervous System Diseases–Alzheimer’s Disease cohort [19]´ In the reference 19, 4309 participants were enrolled in a longitudinal study from 2018–2019. In the present work, the prospective investigation was initiated in 2017. Are the same participants enrolled in both studies? Is this a follow-up study?

´Participants were categorized by BMI into two groups: < 24.0 kg/m² and ≥ 24.0 141 kg/m² [27].´ This criteria could be explained to facilitate the understanding of the threshold.

Author Response

Comment 1: As the authors stated in the limitations of the work, spite the population is not fully representative of the overall region with this diet pattern, the amount of individuals evaluated allow to provide solid conclusions. In this context, the conclusions of the work must be further elaborated.

Response: Thank you very much for your suggestions. The conclusion has been revised.
Comment 2: The section Study protocol must be further detailed/clarified. ´Our study utilized the follow-up evaluation data from the Community-based Cohort Study on Nervous System Diseases–Alzheimer’s Disease cohort [19]´ In the reference 19, 4309 participants were enrolled in a longitudinal study from 2018–2019. In the present work, the prospective investigation was initiated in 2017. Are the same participants enrolled in both studies? Is this a follow-up study?

Response: Thank you very much for your suggestions. The Community-based Cohort Study on Nervous System Diseases–Alzheimer’s Disease Cohort conducted its research across 4 selected survey sites nationwide, with Zhejiang Province being one of these sites. The project launched its baseline survey in 2018 and conducted the first follow-up survey in 2020.

Comment 3: ´Participants were categorized by BMI into two groups: < 24.0 kg/m² and ≥ 24.0 141 kg/m² [27].´ This criteria could be explained to facilitate the understanding of the threshold.

Response: Thank you very much for your suggestions. Previous studies indicated that overweight and obesity may be associated with cognitive impairment. According to the BMI criteria, a value of ≥ 24.0 kg/m² is defined as overweight or obesity. Therefore, participants were categorized by BMI into two groups: < 24.0 kg/m² and ≥ 24.0 kg/m².

Round 2

Reviewer 1 Report

Comments and Suggestions for Authors

The authors have positively revised the manuscript, and my concerns were properly addressed. So, I can recommend this manuscript for publication.

Reviewer 3 Report

Comments and Suggestions for Authors

The authors modified the text according to suggestions. The manuscript has been sufficiently improved to warrant publication.